# Constructing temporal networks with bursty activity patterns

Anzhi Sheng ®[1,2], Qi Su[3,4,5], Aming Li ®[1,6] ✉, Long Wang ®[1,6] ✉ & Joshua B. Plotkin ®[2,7] ✉

Human social interactions tend to vary in intensity over time, whether they are in person or online. Variable rates of interaction in structured populations can be described by networks with the time-varying activity of links and nodes. One of the key statistics to summarize temporal patterns is the inter-event time, namely the duration between successive pairwise interactions. Empirical studies have found inter-event time distributions that are heavy-tailed, for both physical and digital interactions. But it is difficult to construct theoretical models of time-varying activity on a network that reproduce the burstiness seen in empirical data. Here we develop a spanning-tree method to construct temporal networks and activity patterns with bursty behavior. Our method ensures any desired target inter-event time distributions for individual nodes and links, provided the distributions fulfill a consistency condition, regardless of whether the underlying topology is static or time-varying. We show that this model can reproduce burstiness found in empirical datasets, and so it may serve as a basis for studying dynamic processes in real-world bursty interactions.

Temporal networks have been recognized as a powerful tool to model complex systems with time-varying interactions[1–3]. A large body of literature concentrates on analyzing the activation dynamics of nodes and links in such networks. The inter-event time (the waiting time between two consecutive interaction events, IET) is a canonical measure of temporal patterns, and it is known to have profound effects on individual behavior[4–7] and dynamical processes occurring on networks[8–12]. A variety of empirical datasets, such as email and mobile communications[13–15], epidemic transmission[16–18], and human mobility[19,20], exhibit non-Poisson activity patterns, known as burstiness[4,21,22]. These IET patterns are characterized by periods of frequent activation interleaved with long periods of silence. Empirical networks often exhibit burstiness in both the activity of individuals (nodes) as well as interactions (edges)[23–25].

It has proven difficult to construct synthetic temporal networks whose properties are similar to the bursty behavior seen in empirical temporal networks[26]. Previous approaches can be divided into two categories: structure-based modeling, and contact-based modeling. The former approach applies dynamical processes to static underlying topologies, such as random walks[27,28], link dynamics[29], and inhomogeneous Poisson processes[30]. For example, Barrat et al. used random itineraries on weighted underlying networks to generate time-extended structures with bursty behavior[28]. The latter approach uses a stream of contacts generated by certain realistic mechanisms, such as social appeal[31], individual resource[32], and memory[33]. For example, Perra et al. proposed the activity-driven model[34], in which each node, isolated in the beginning, becomes active with a probability proportional to its own activity potential and forms links with other nodes.

[1]Center for Systems and Control, College of Engineering, Peking University, Beijing 100871, China. [2]Department of Biology, University of Pennsylvania, Philadelphia, PA 19104, USA. [3]Department of Automation, Shanghai Jiao Tong University, Shanghai 200240, China. [4]Key Laboratory of System Control and Information Processing, Ministry of Education of China, Shanghai 200240, China. [5]Shanghai Engineering Research Center of Intelligent Control and Management, Shanghai 200240, China. [6]Center for Multi-Agent Research, Institute for Artificial Intelligence, Peking University, Beijing 100871, China. [7]Center for Mathematical Biology, University of Pennsylvania, Philadelphia, PA 19014, USA. ✉e-mail: amingli@pku.edu.cn; longwang@pku.edu.cn; jplotkin@sas.upenn.edu

Several related models have been proposed based on the activity-driven framework[35,36]. Despite a large body of studies in constructing temporal networks, they usually fail to reproduce the same level of burstiness as empirical datasets, and they lack mathematical guarantees for the behavior of the synthetic network. These goals require a model to generate bursty behavior simultaneously in the activity of both nodes and links, while the level of burstiness in nodes and links is allowed to be easily modified in a reasonable parameter space.

In this study, we provide an analytical framework to systematically construct temporal networks on both static and time-varying underlying topologies. Our construction algorithm can reproduce the burstiness of both nodes and edges in four empirical datasets, including social interactions in rural Malawi, colleague relationships in an office building over two years, and friendship relations in a high school. The assumptions of our model can also be tested in the empirical datasets. Our construction thus serves as an efficient method to generate realistic temporal networks that can then be used to investigate dynamical processes (such as evolutionary dynamics, social contagion, or epidemics) on temporal networks.

Our approach to constructing temporal networks uses a spanning-tree method. Spanning trees are widely recognized as an important family of sparse sub-graphs since they tend to govern dynamical processes on full graphs[37–40]. For example, in social networks, the backbone of an aggregated communication topology is often constructed as the union of shortest-path spanning trees, on which information flows fastest[39]. As a result, a large portion of directed edges are bypassed by faster indirect routes in the tree. In studies of food webs[40], spanning trees are defined as the flows from the environment to every species. The links in or out of the tree are denoted as 'strong' or 'weak' links, related to delivery efficiency or system robustness and stability.

## Results

We introduce a spanning-tree method for constructing temporal networks on any underlying topology, which restricts the interaction pattern between pairs of individuals. The activity of every single node and link is a binary-state process, switching between active and inactive. We use inter-event time (IET) distribution, which measures the time intervals between consecutive activations, to quantify the activity patterns of nodes and edges.

Our method allows for both static and time-varying underlying topologies. The underlying topology represents physical limitations on pairwise interactions, where an interaction between two individuals is possible only when there is a link between them in the underlying topology. Here we first consider static underlying topologies, in which the activation dynamics on a given topology is much faster than the evolution of the underlying topologies, such as the time-varying traffic flow on a relatively stable road network. We consider three classes of topologies in order of increasing complexity: two-node topologies, tree topologies, and finally arbitrary structured topologies. After studying static topologies, we will subsequently consider cases when the underlying topology itself also changes over time.

### Two-node systems

We begin with a basic unit of a networked system – a two-node system, which is composed of nodes $x$ and $y$ and a link $z$ between them (see Fig. 1a). The nodes and edges are either active or inactive at each discrete time step. We assume that the state updating of $x$ follows a

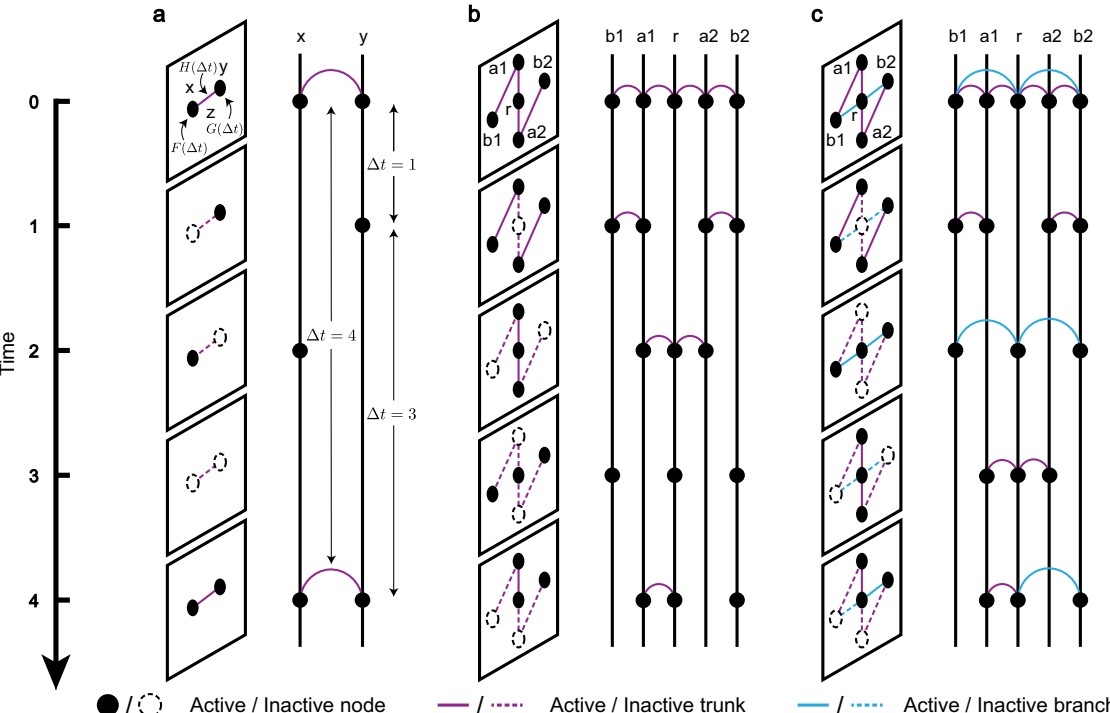

**Fig. 1 | Schematic illustration of constructing temporal networks on different underlying topologies.** Each node/link switches between two states, active (solid circle/line) and inactive (dashed circle/line), and all nodes and links are set to be active initially. **a** The basic unit of a network system is a two-node system with nodes $x$, $y$, and a link $z$ connecting them. At the beginning of temporal network construction, three probability mass functions $F(\Delta t)$, $G(\Delta t)$, $H(\Delta t)$ are given as the expected IET distributions for $x, y, z$, where $\Delta t$ represents the time interval between two consecutive activations. Then, the activity of $x, y, z$ is driven by the renewal processes with the corresponding expected IET distributions. We constrain that $z$ is active only if both $x$ and $y$ are active. **b** An extension of two-node systems is tree systems, in which nodes are divided into two categories, a root ($r$) and leaves ($a1$, $a2$, $b1$, $b2$). The state of nodes and links is updated by sequentially executing the algorithm over each two-node system from the root ($r$-$a1$ and $r$-$a2$) to the outermost leaves ($a1$-$b1$ and $a2$-$b2$). **c** There is at least one spanning tree for any static underlying topology. The links in the spanning tree are called trunks (purple lines) and the links outside the spanning tree are called branches (blue lines). The states of nodes and trunks are updated first, and then the states of branches are established according to the state of the nodes on both sides.

renewal process $\{X_n\}_{n \geq 0}$ and assigns $x$ a probability mass function $F(\Delta t)$ as its target IET distribution (see Supplementary Information section 1). The random variable $X_n$ equals 1 if $x$ is active at time $n$, otherwise $X_n = 0$. Likewise for node $y$ and edge $z$, which have respective target IET distributions $G(\Delta t)$ and $H(\Delta t)$, and respective renewal processes $\{Y_n\}_{n \geq 0}$ and $\{Z_n\}_{n \geq 0}$. The initial state of $x, y, z$ is active (i.e., $X_0 = Y_0 = Z_0 = 1$). The goal is to construct a two-node temporal network that satisfies the target IET distributions of $x$, $y$, and $z$.

By definition, we say that edge $z$ is active when $x, y$ are both active (i.e., $Z_n = X_n Y_n$). Furthermore, we assume that, given the trajectory of $x$ until $n$, the probability of $x$ being active at time $n + 1$ is independent of the trajectory of $y$ until $n$, that is,

$$\mathbb{P}(X_{n+1}, \mathbf{Y}^{(n)} | \mathbf{X}^{(n)}) = \mathbb{P}(X_{n+1} | \mathbf{X}^{(n)}) \cdot \mathbb{P}(\mathbf{Y}^{(n)} | \mathbf{X}^{(n)}), \qquad (1)$$

where $\mathbf{X}^{(n)}$ is a random vector of length $n + 1$, equal to $(X_0, X_1, \ldots, X_n)^{\mathrm{T}}$. The vector $\mathbf{X}^{(n)}$ records the whole history of states of $x$ through time $n$, and so does $\mathbf{Y}^{(n)}$. Given these assumptions, we can show that when all targeted distributions are identical (i.e., $F = G = H$) the system must be completely synchronous, that is, all of $x, y, z$ are either active or inactive at each time step (see Supplementary Information section 2).

Given the state of the first $n$ times (from time 0 to $n - 1$), the conditional probability that node $x$ is active at time $n$ is given by

$$\mathbb{P}(X_n = 1 | X_{n-1} = w_x^{(n-1)}, \ldots, X_0 = w_x^{(0)}) := p_x(\mathbf{w}_x^{(n-1)}, 1) = \frac{F(n - m)}{\sum_{i \geq n - m} F(i)}. \quad (2)$$

where $\mathbf{w}_x^{(n-1)} = (w_x^{(0)}, \ldots, w_x^{(n-1)})^{\mathrm{T}} \in \{0,1\}^n$ records all historical states of node $x$ before time $n$, called $x$'s trajectory, and $m = \max\{k \leq n : w_x^{(k)} = 1\}$ represents the last activation time of $x$. Analogous conditional probabilities apply to $y$ and $z$, and it is straightforward to show that each element of $\mathbf{w}_z^{(n-1)}$ equals the product of the corresponding elements of $\mathbf{w}_x^{(n-1)}$ and $\mathbf{w}_y^{(n-1)}$.

We propose an algorithm to construct two-node temporal networks such that the IET distributions of $x, y, z$ will match the desired targeted distributions $F, G, H$ and the desired total time duration $t_{\text{tol}}$. At each time step $t = n + 1$ ($0 \leq n \leq t_{\text{tol}} - 1$), we calculate four probabilities with Eq. (2),

$$
\begin{aligned}
p_1 &= \mathbb{P}(X_{n+1} = 1, Y_{n+1} = 1 | \mathbf{X}^{(n)}, \mathbf{Y}^{(n)}) = p_z(\mathbf{w}_z^{(n)}, 1), \\
p_2 &= \mathbb{P}(X_{n+1} = 1, Y_{n+1} = 0 | \mathbf{X}^{(n)}, \mathbf{Y}^{(n)}) = p_x(\mathbf{w}_x^{(n)}, 1) - p_z(\mathbf{w}_z^{(n)}, 1), \\
p_3 &= \mathbb{P}(X_{n+1} = 0, Y_{n+1} = 1 | \mathbf{X}^{(n)}, \mathbf{Y}^{(n)}) = p_y(\mathbf{w}_y^{(n)}, 1) - p_z(\mathbf{w}_z^{(n)}, 1), \\
p_4 &= \mathbb{P}(X_{n+1} = 0, Y_{n+1} = 0 | \mathbf{X}^{(n)}, \mathbf{Y}^{(n)}) = 1 + p_z(\mathbf{w}_z^{(n)}, 1) - p_x(\mathbf{w}_x^{(n)}, 1) - p_y(\mathbf{w}_y^{(n)}, 1).
\end{aligned}
$$
$$(3)$$

These four probabilities represent the conditional probabilities for the four possible states of nodes $x$ and $y$, given the previous states of nodes $x$ and $y$ (i.e., $\mathbf{w}_x^{(n-1)}$ and $\mathbf{w}_y^{(n-1)}$). Specifically, $p_1$ represents the probability that edge $z$ is active at time $t = n + 1$; $p_2$ (or $p_3$) represents the probability that $z, y$ are inactive but $x$ is active; $p_3$ represents the probability that $z, x$ are inactive but $y$ is active; and $p_4$ represents the probability that $x, y, z$ are all inactive at time $t$. Next, we determine the state of $x$ and $y$ at $t$, in order. The probability that $x$ is active is $p_x(\mathbf{w}_x^{(n)}, 1)$. If $x$ is active, the probability of $y$ being active is $p_1 / p_x(\mathbf{w}_x^{(n)}, 1)$. Otherwise, the probability becomes $p_3 / (1 - p_x(\mathbf{w}_x^{(n)}, 1))$. Finally, we update the trajectory of $z$ by the relation $w_z^{(t)} = w_x^{(t)} w_y^{(t)}$. The construction stops when $t = t_{\text{tol}}$. Algorithm 1 in Supplementary Information outlines the above procedure. It is worth noting that the activation order of $x$ and $y$ does not affect the IET distributions of $x, y, z$.

Although the algorithm is specified for an arbitrary combination of target IET distributions ($F, G, H$), these distributions must satisfy an implicit condition in order to guarantee the consistency of the construction. For example, if the target distributions specify that the link $z$ is activated more frequently than the nodes $x$ and $y$, then no construction

is possible, because the link is active only when both nodes are active. This would result in at least one of the probabilities $p_i$ ($i = 1, 2, 3, 4$) in Eq. (3) being less than 0 during the construction. We say that the combination ($F, G, H$) of target distributions is consistent if $p_i$ ($i = 1, 2, 3, 4$) belong to [0, 1] for all possible trajectories $\mathbf{w}_x^{(n)}$ and $\mathbf{w}_y^{(n)}$ with any length $n$. When a two-node system is consistent, the algorithm is well-defined, and it ensures that the IET distributions of $x, y, z$ will satisfy the targets $F, G, H$, respectively (see Supplementary Information section 2).

## Tree systems

The construction for two-node systems can be naturally extended to tree systems, which consist of a number of interconnected two-node systems (Fig. 1b). We randomly select a node as the root $r$ and classify the remaining nodes (i.e., leaves) according to their distance from $r$. We choose a desired target IET distribution for each node and link. At each time step, we first determine the state of $r$, which is only related to its own trajectory. Then, every leaf one step away from $r$ ($a1$ and $a2$ in Fig. 1b) forms a two-node system with $r$, and the states of these leaves are determined by Algorithm 1. Next, all leaves one step away from $r$ form two-node systems with their corresponding leaves two steps away from $r$ [($a1$, $b1$) and ($a2$, $b2$) in Fig. 1b]. Analogously, the states of all leaves are updated within a two-node system in order of their distance from $r$. Algorithm 2 in Supplementary Information summarizes this procedure.

This procedure requires an additional assumption of conditional independence – that for a pair of two-node systems sharing a node, if the state of the common node is given then the activity of the other two nodes is independent. As shown in Fig. 1b, we present two examples, ($r, a1, a2$) and ($r, a1, b1$), fulfilling this condition. The former indicates that the activity of nodes with the same distance from $r$ (i.e., $a1, a2$) is independent given the state of their common node (i.e., $r$), which is closer to the root. The latter indicates that the activity of a node (i.e., $b1$) is not affected by the node that is more than one step away (i.e., $r$), given the state of the intervening node (i.e., $a1$).

It is straightforward to show that a tree system is consistent if all two-node systems in the tree are consistent. When a tree system is consistent, the IET distribution of every single node/link generated by this construction is guaranteed to match its corresponding target distribution (see Supplementary Information section 2).

### Spanning-tree based construction

For any static (but arbitrary) underlying topology, we can always find a spanning tree (Fig. 1c). A link is called a trunk if it is in the spanning tree, and it is called a branch otherwise. In our construction, we first randomly select a spanning tree and the root of the tree, and then choose a target IET distribution for each node and trunk. Next, we execute Algorithm 2 on the spanning tree, so that all nodes and trunks will update their states to achieve the targeted IETs. The state of each branch is then active when the nodes on both ends are active. As a result, the activity of the entire system is determined by its spanning tree. Algorithm 3 in Supplementary Information summarizes this procedure. We say that the system is consistent when its spanning tree system is consistent.

### Synthetic temporal networks

To test our algorithm, we construct temporal networks with one of two activity patterns—bursty activity patterns and Poisson activity patterns. Bursty patterns arise when there is a simultaneous bursty activity in node and links activity, and Poisson pattern arises in settings such as bank queuing systems[41] and spreading dynamics[42]. In particular, we choose target IET distributions of nodes and trunks that are power-law distributions for bursty activity patterns, given by

$$p(\Delta t; \alpha) \sim \Delta t^{-\alpha} \quad (\alpha > 1). \qquad (4)$$

And we choose discrete exponential distributions for Poisson-like activity patterns, given by

$$p(\Delta t; \alpha) \sim \int_{\Delta t - 1/2}^{\Delta t + 1/2} \alpha e^{-\alpha x} dx \quad (\alpha > 0), \quad (5)$$

where $\Delta t$ represents the inter-event time and $\alpha$ the exponent. The respective survival functions are given as

$$\mathbb{P}(T > \Delta t) \sim \Delta t^{-\alpha + 1} \quad (6)$$

with exponent $\alpha - 1$ and

$$\mathbb{P}(T > \Delta t) \sim e^{-\alpha \Delta t} \quad (7)$$

with exponent $\alpha$.

In the main text, we focus on a simple case when all nodes (trunks) have a common exponent $\alpha_{pmf}$ ($\beta_{pmf}$), and we use the aggregated IET distribution[21-23], which counts the IETs of all nodes or links, to quantify the intensity of activity. In Supplementary Information, we also consider the IET distributions of every single node and link (Supplementary Fig. 1) and we investigate the relationship with the aggregated IET distributions. We also explore a more general case in which the exponents of nodes and trunks are sampled independently from a distribution (Supplementary Fig. 2).

We begin our analysis by constructing temporal networks with bursty activity patterns. We derive a necessary and sufficient condition for system consistency, which applies to any network length $t_{tol}$ (see Methods Eq. (14) and Supplementary Information for details). As examples, we consider two pairs of exponent setups, $(\alpha_{pmf}, \beta_{pmf}) = (2.00, 1.90)$ and $(\alpha_{pmf}, \beta_{pmf}) = (1.80, 1.30)$, and we execute Algorithm 3 over two classes of static underlying topologies, Barabási-Albert scale-free networks[43] and Watts-Strogatz small-world networks[44]. Figure 2a shows the aggregated IET distributions of nodes and links. For both underlying topologies and both parameter setups, the probability mass functions $\mathbb{P}(\Delta t)$ and the corresponding survival

functions $\mathbb{P}(T > \Delta t)$ are well fit by power-law distributions, showing simultaneous burstiness in nodes and links. The best-fit exponent for nodes matches the exponent of the target IET distribution, and the exponent for links is slightly lower than the target exponent, due to the impact of branches (edges outside the spanning tree, where the algorithm is guaranteed to work).

Next, we construct temporal networks with Poisson-like activity patterns. If the target distributions for nodes and trunks are Poisson distributions, then we can prove that the system is never consistent (see Supplementary Information section 2). However, it is possible to construct consistent systems when the target IET follows discrete exponential distributions. In this case, we derive a necessary and sufficient condition for system consistency—namely, that the difference of $\alpha_{pmf}$ and $\beta_{pmf}$ lies in [0, ln 2], meaning that the activity of nodes must be more frequent than links but not too frequent (see Methods and Supplementary Information for details). Figure 2b shows the aggregated IET distributions along with the exponents of the target distributions $(\alpha_{pmf}, \beta_{pmf}) = (2.50, 2.00)$ and $(1.80, 1.30)$. All the distributions follow the expected exponential decay.

Comparing the results for the two topologies, we find that the exponents produced by the algorithmic construction are sensitive to the choice of target IET distribution, but are stable to the choice of network topology. To examine this observation more generally, we investigated a wide range of random regular underlying topologies with different average degrees, ranging from 5 to the well-mixed case (i.e., each node linked to all other nodes, see Supplementary Figs. 3 and 4). We find that we can robustly match target IET distributions across all these topologies: the relative deviation between the largest and smallest exponent in algorithmically constructed networks is within 4%.

We can understand why the algorithmic construction for producing a desired IET distribution is not significantly dependent on topology by analyzing the activity of branches. In particular, we prove that the IET distribution of every single branch is approximately a power-law distribution (respectively a discrete exponential distribution) in bursty activity patterns (respectively Poisson-like activity patterns) with uniform upper and lower bounds related only to the target

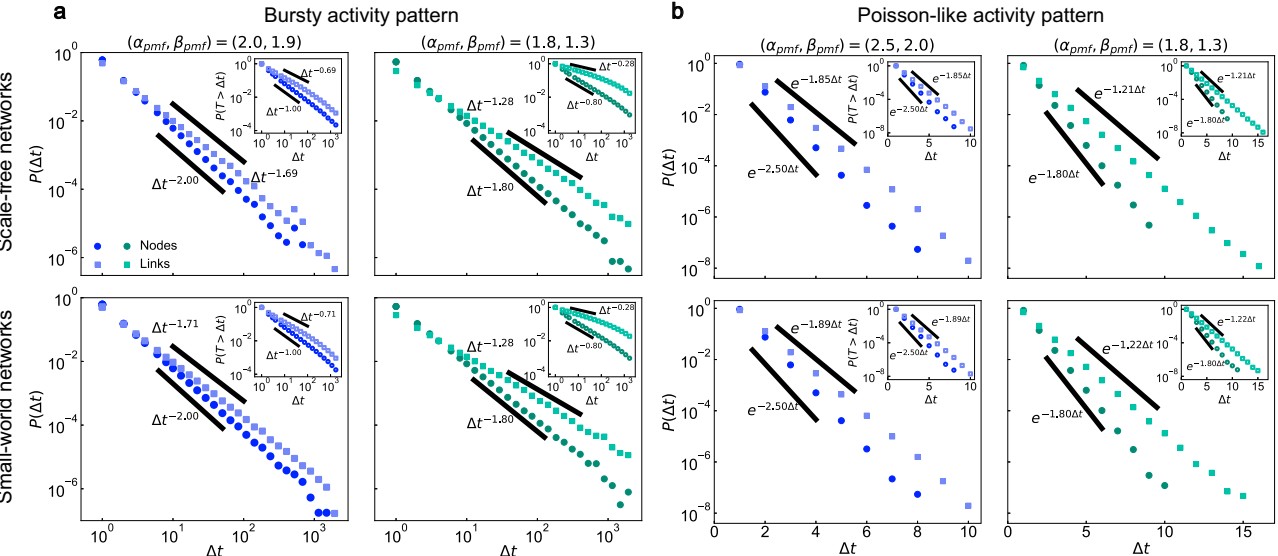

**Fig. 2 | Aggregated IET distributions of nodes and links on different underlying topologies.** We consider the construction of temporal networks on two classes of static underlying topologies, Barabási-Albert scale-free networks (first row) and Watts-Strogatz small-world networks (second row). The targeted exponent of every single node (trunk) is identical, denoted as $\alpha_{pmf}$ ($\beta_{pmf}$). We select a pair of high exponents (blue dots) and a pair of low exponents (green dots) for the bursty activity pattern (**a**) and the Poisson-like activity pattern (**b**), given as ($\alpha_{pmf}$,

$\beta_{pmf}$) = (2.00, 1.90), (1.80, 1.30) and ($\alpha_{pmf}$, $\beta_{pmf}$) = (2.50, 2.00), (1.80, 1.30), respectively. The algorithmic distributions of nodes (circles) and links (squares) are well-predicted by power-law distributions in (**a**) and by exponential distributions in **b**. The thick black lines with fitted exponents are plotted for reference. The distributions are the average over 50 independent trials. Parameter settings: network size $N = 10^3$, average degree $k = 6$, and network length $t_{tol} = 10^4$ in **a** and $t_{tol} = 10^3$ in **b**.

distributions (see Supplementary Information section 3). This is also why the performance of our method is stable with respect to the choice of spanning tree (Supplementary Fig. 5).

## Empirical temporal networks

We tested the ability of our algorithm to reproduce the burstiness of activity patterns observed in four empirical datasets, collected by the SocioPatterns collaboration. These four datasets record pairs of face-to-face interactions from different social contexts, ranging from a village in rural Malawi, to an office building and a high school in France. Each dataset is comprised of contact events with timestamps, represented by triplets $(t, i, j)$–indicating the occurrence of an interaction between individual $i$ and individual $j$ at time $t$.

It is worth noting that the empirical data record only the communication moments, so that only the active nodes with at least one active neighbor can be detected. In other words, the empirical data are observations of the inter-communication times (ICTs), rather than the IETs of nodes. Nevertheless, we demonstrate that the ICT distribution of single nodes converges exponentially to the IET distribution as the number of neighbors on the underlying topology increases (see Supplementary Information section 4). Since empirical datasets often originate from highly connected populations, the ICT distributions approximate the statistical properties of the corresponding IET distributions.

Before applying our algorithm, we first test whether the assumptions underlying the algorithmic construction are consistent with the empirical datasets. The assumption that the activity of nodes and links is a renewal process is reasonable, compared to the empirical data (Supplementary Fig. 6). However, the strong form of the conditional independence (Eq. (1)) assumed by our construction is rejected for the empirical data (Supplementary Fig. 7). Nonetheless, the empirical data satisfy a weaker form of conditional independence (Supplementary Fig. 8, see Supplementary Information section 5 for details).

After pre-processing the datasets (see Methods), we obtain four empirical temporal networks with population sizes ranging from $N = 84$ to $N = 327$ and length from $t_{tol} = 7,375$ to $t_{tol} = 43,436$ time-steps. For all of these temporal networks, the empirical ICT distributions of nodes and links both exhibit heavy tails, with different decay rates, showing simultaneous burstiness in activity. We fit these empirical ICTs with power-law distribution (Eq. (4)) by maximum likelihood estimation[45,46], and we use the fitted distributions as targets for constructing synthetic temporal networks. Figure 3 shows the comparison between the empirical and algorithmically constructed ICT distributions of nodes and links. Our algorithm successfully replicates the qualitative patterns of burstiness observed in empirical datasets. Supplementary Fig. 9 shows the comparison between the IET and ICT distributions of nodes. Since the average degree of these empirical underlying topologies is large, the IET distribution collapses onto the ICT distribution.

## Combination with network evolution

Although some underlying topologies are static, a variety of real-world systems also exhibit topology changes over timescales that are comparable to the activation dynamics on the network. For example, in online social networks, new users can enter the network and engage in new interactions with existing users; or users can switch between online and offline states. As a consequence, temporal changes in activity originate not only from the states of existing nodes and links, but also from the addition and subtraction of nodes and links in the underlying structure. In these cases, the underlying topology (i.e., physical limitations on interactions) is no longer static, but time-varying.

With this as a backdrop, we extend our algorithm from static to dynamic underlying topologies, starting first with networks that grow in size. We introduce a new model that combines our algorithm for constructing temporal networks with the Barabási-Albert model[43], which we call the temporal Barabási-Albert model. The construction process is as follows. An underlying topology is initialized with $m_0$ nodes, and the spanning tree is selected randomly. At each time step,

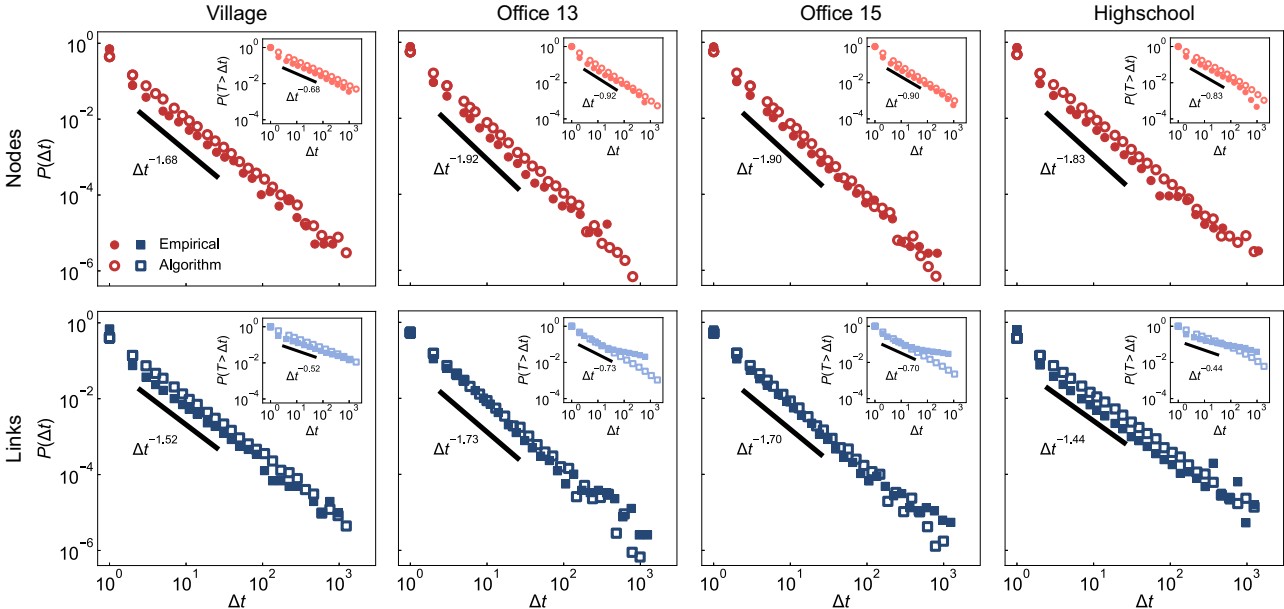

**Fig. 3 | Burstiness in empirical datasets.** We analyze four empirical temporal networks of social interactions within and across households among 84 individuals in a village[69]; colleague relationships among 95 and 219 employees in an office building in two different years (2013 and 2015)[25,70]; and friendship and educational relationships among 327 students in a high school in Marseilles[71]. The length of these temporal networks from left to right are 43436, 20129, 21536, and 7375. For each empirical temporal network, we count the aggregated ICT distributions of nodes (solid circles) and links (solid squares), which both present bursty behaviors and are well-predicted by power-law distributions. The thick black lines are power-law distributions with the fitted exponents. We take the fitted distributions as targets and obtain the respective algorithmic distributions of nodes (hollow circles) and links (hollow squares). The algorithmic distributions present the same level of burstiness as the corresponding empirical distributions.

the activity state of the existing network updates once with Algorithm 3, then one adds a new node with $m \leq m_0$ links connected to $m$ different existing nodes following the preferential attachment rule[43]. All the newly added elements are set to be active, and the states of the older pre-existing elements are updated accordingly. The spanning tree is then updated by adding the new node and a link randomly selected from $m$ new links. At some time point, the underlying topology stops growing, and the construction process continues with $g \geq 0$ more steps on the final state of the underlying topology. Figure 4a shows a schematic illustration of the above procedure. Figure 4b shows numerical simulations of the temporal Barabási-Albert model in bursty and Poisson-like activity patterns. We find that the IET distributions of nodes and links are stable with respect to the duration time $g$, which means that the activity pattern is established during the evolution of the underlying topology, and it is then preserved after the topology is fixed.

Aside from overall growth in system size, some systems change underlying topologies without increasing in size, but rather by changes in the links between individuals, even if the total number of links is stable over time. We show that our model is also applicable in this case (see Supplementary Figs. 10 and 11). In addition to changing the location of edges, nodes, and links may also be removed over time, due to aging or other recessionary impacts, which is a counterpart of system growth; a straightforward example is the reverse process of the temporal Barabási-Albert model.

To model these various kinds of network evolution, we have developed a more general procedure for constructing temporal networks on time-varying underlying topologies (see Supplementary Information Algorithm 4). The key to this algorithm is to update the spanning tree in accordance with the evolution of the underlying topology.

## Topology of aggregated static networks

In addition to bursty in inter-event times, real-world temporal networks exhibit several other characteristic and distinguishing features[28,47]. One important feature, distinct from the IET distribution, is the topology of the weighted network generated by integrating

snapshots of activity over time. In particular, we can consider the weighted network produced by aggregating active nodes and edges from time $t = 1$ to time $t = t_{agg}$ (Fig. 5a). The aggregated network is weighted, and we characterize its topology by considering the distribution of weighted node degree, which is a generalization of the node degree distribution.

Our algorithm for constructing a temporal network with desired IET distributions also can reproduce the stationary property of the associated weighted node degree distribution. In particular, if the node degree distribution of the (unweighted) static underlying topology follows $d(x)$, and if all links have the same activity pattern specified by parameter $\beta$, then for a sufficiently large aggregation time $t_{agg}$ we can prove that the weighted node degree distribution produced by our algorithm will satisfy

$$p_{t_{agg}}(x) \sim \frac{v(\beta)d(xv(\beta)/t_{agg})}{t_{agg}}. \tag{8}$$

Here $v(\beta)$ is a function of the parameter $\beta$ that governs activity patterns (see Supplementary Information section 7 for detailed derivations). Equation (8) shows that, when the underlying topology is a scale-free network (i.e., $d(x)$ follows a power-law distribution), then $p_{t_{agg}}(x)$ will also follow a power-law distribution with the same exponent as $d(x)$, regardless of the aggregation duration $t_{agg}$. Figure 5b shows an example of this general mathematical result in the case of a Barabási-Albert scale-free underlying topology.

For an arbitrary (unweighted) node degree distribution $d(x)$, we can still find a normalization method such that the aggregated distribution is insensitive to the duration $t_{agg}$. The normalization proceeds as follows: First, an aggregated network of aggregation time $t_{base}$ is selected as the baseline. Then for any aggregated network of aggregation time $t_{agg} > t_{base}$, we multiply $t_{base}/t_{agg}$ by each node strength. When $t_{base}$ is sufficiently large, the normalized survival function for any $t_{agg} > t_{base}$ will collapse onto the survival function for $t_{base}$ (see Supplementary Information section 7 for detailed derivations). We have verified this result on a network with a small-world topology (Fig. 5c).

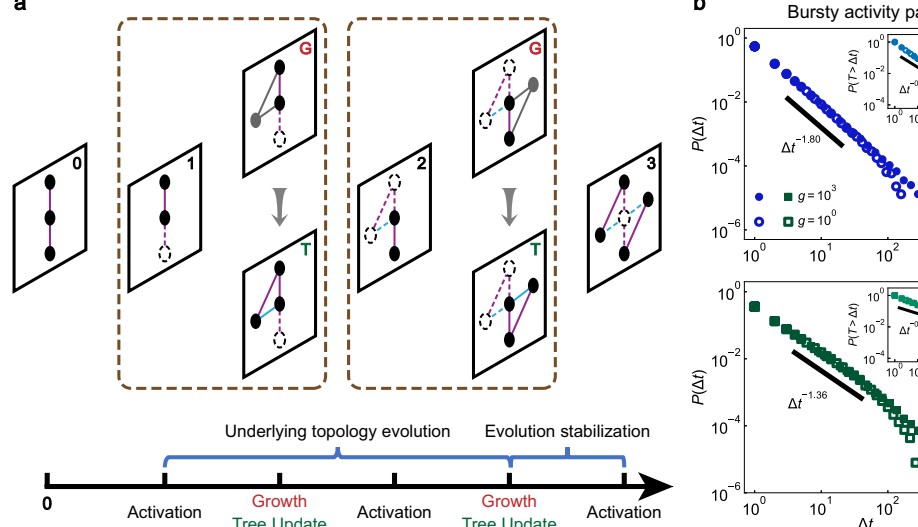
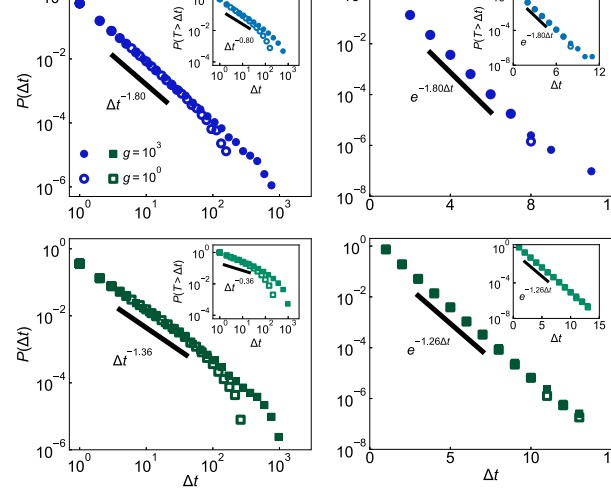

**Fig. 4 | Construction on time-varying underlying topologies. a** We consider the temporal network construction with a time-varying topology modeled by the Barabási-Albert model. There are $m_0$ nodes in the initial snapshot. When the underlying topology is evolving, a node with $m$ link(s) enters the network system (snapshot G), and the spanning tree is updated accordingly (snapshot T). When the evolution is stable, the construction process degenerates to that for the static underlying topology. **b** We consider the impact of the duration time $g$ after network evolution stabilization. Our model produces expected bursty and Poisson-like activity patterns on such a time-varying underlying topology. Furthermore, the activity patterns formed during network evolution (hollow dots) are maintained after evolution stabilization (solid dots). Parameter settings: $\alpha_{pmf} = 1.8$, $\beta_{pmf} = 1.5$, $m_0 = 3$, $m = 3$, and final network size $N = 500$.

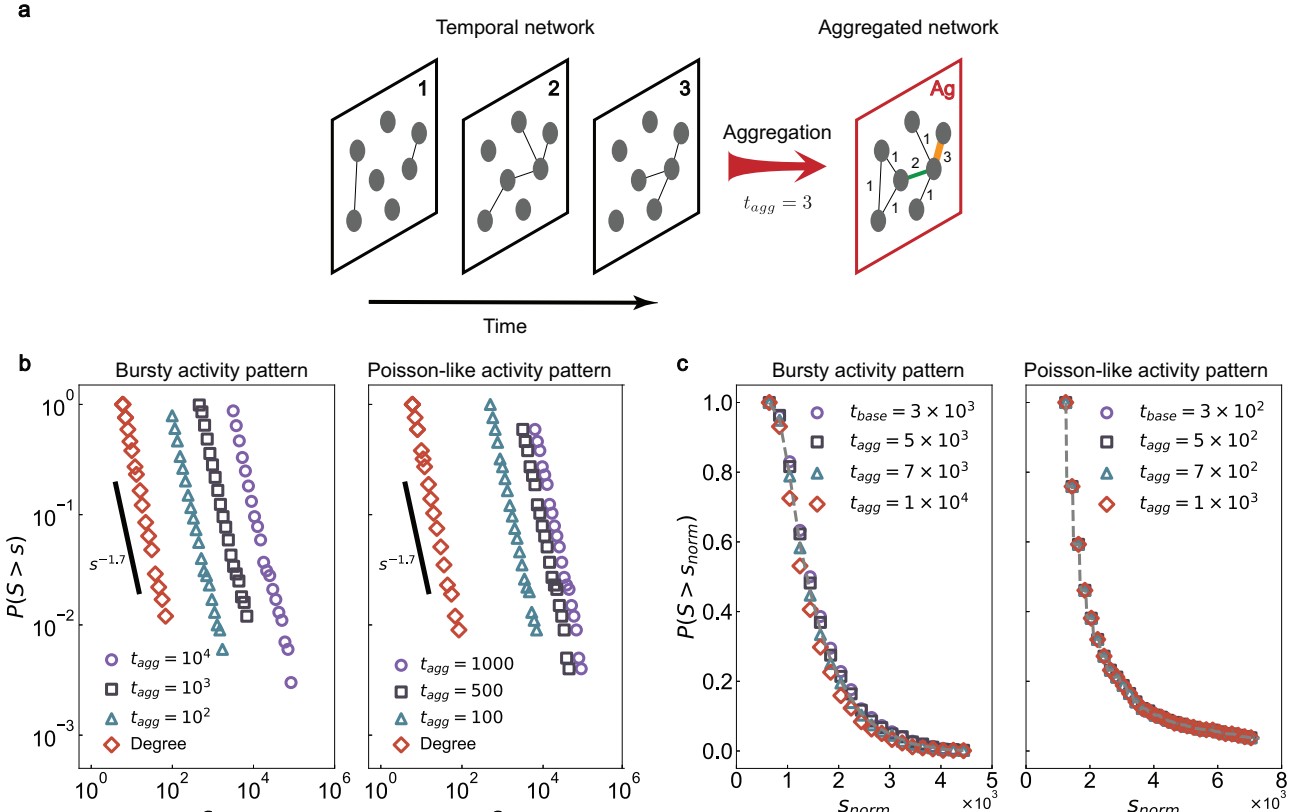

**Fig. 5 | Node strength of aggregated temporal networks. a** The aggregated network is a weighted network (network Ag) that collates all interactions from the first snapshot, at time $t = 1$, though the snapshot at time $t = t_{agg}$. Link weights represent the activation numbers summed over network evolution, which can exceed 1 (such as the orange and green links). **b, c** The stationary property of node strength distributions on synthetic temporal networks. We consider two (unweighted) underlying topologies: Barabási-Albert scale-free networks (**b**) and Watts-Strogatz small-world networks (**c**). For the first topology, the survival function of the degree distribution follows a power-law distribution with an exponent

$\gamma \approx 1.7$ (red diamonds). The survival function of the node strength for aggregated networks also follows a power-law distribution with the same exponent as for the underlying topology, regardless of the aggregation duration $t_{agg}$. For the second topology, we set a baseline aggregated network of aggregation time $t_{base}$. For any aggregation duration $t_{agg} > t_{base}$, we normalize the survival function of node strength with respect to $t_{base}$, and we find that the survival function of the aggregated network collapse onto that of the baseline network, regardless of the duration $t_{agg}$. Parameters: $(\alpha_{pmf}, \beta_{pmf}) = (2.00, 1.90)$ for the bursty activity pattern and $(\alpha_{pmf}, \beta_{pmf}) = (2.50, 2.00)$ for the Poisson-like activity pattern.

Finally, we have applied this normalization procedure to the aggregated activity patterns of the four empirical datasets shown in Fig. 3. This analysis verifies that our construction of temporal networks to match empirical IET distributions also matches the stationary property of the weighted node degree in empirical data (Supplementary Fig. 12).

## Discussion

Simple models that neglect temporal variation in individual behavior do not suffice to describe the dynamics of many real-life complex systems. A large and growing body of studies suggest that state switching of individuals and interactions plays a significant role in diverse dynamical processes, such as face-to-face communication[25], evolutionary dynamics[12,48], and network control[49]. We have proposed an analytical framework and corresponding spanning-tree method to construct temporal networks with specific activity patterns, including bursty and Poisson-like activity patterns. Unlike prior constructions of temporal networks in the literature[5,35,36], our algorithm is able to reproduce the simultaneous burstiness of both nodes and edges observed in empirical datasets, from diverse social contexts (Supplementary Fig. 13).

The central ingredient in our construction algorithm—the spanning tree—has been widely recognized as a significant feature in both theoretical[50–52] and real-world applications of network science[53]. For example, in path-finding algorithms such as Dijkstra's algorithm[54] and

the A* search algorithm[55], the shortest paths from a given source node to all other nodes, together with the source node, form a shortest-path tree. These algorithms are widely used in mobile robot covering problems[56], for tracking the establishment of oil pipelines[57], and for vehicle routing[58]. The tree structure is the backbone of these networked systems. Other examples include telecommunication networks, including the Internet, where the Spanning Tree Protocol[59] and Augmented tree-based routing[60] are used to avoid routing loops, to solve the scalability problem, and gain resilience against node failure and link instability. In social networks, spanning tree-based algorithms have been proven effective in detecting communities, one of the most widely studied issues in network science[61]. Thus our spanning tree-based method for generating specific activity patterns might have implications in several areas of application, which remain to be investigated.

Our analysis of activity patterns on different underlying topologies shows that the algorithmic IET distribution of nodes or edges does not significantly depend on the underlying topology, for a given spanning tree. This result indicates that the macroscopic activity pattern of a general network can be largely determined by the dynamics of its key components, such as its spanning tree. Branches from the spanning tree will cause only small perturbations to the activity pattern, regardless of their number and location, akin to the effects of 'weak links' in food webs[40]. As a result, we can construct a temporal network with a desired consistent activity pattern, even if the underlying topology is not precisely known, or even changing in time.

Another straightforward way to measure the burstiness and memory of temporal networks is to calculate the burstiness parameter[26] and the auto-correlation function (see Supplementary Information section 8). A larger burstiness parameter means a higher level of burstiness, and a lower absolute value of the auto-correlation function means a weaker dependence on memory. Our results show that our synthetic bursty activity patterns have strong auto-correlation and a positive and high burstiness parameter, while the Poisson-like activity patterns are memoryless and have a negative burstiness parameter (Supplementary Tables 1 and 2, Supplementary Fig. 14).

The past ten years have shown increasing interest in understanding the effects of group interactions and higher-order interactions[62–66], meaning behavioral activities that are not limited to just pairs of individuals. A recent study has shown that higher-order interactions in empirical datasets display similar bursty behaviors to pairwise interactions[67]. And so a natural extension of this work is to study the activity of higher-order interactions in temporal networks. Our approach may provide a method to decompose networks into several elementary components, analogs of spanning trees in the context of hyper-graphs, which remains a direction for future research on the temporal dynamics in groups of interacting agents.

## Methods
### Mathematical formalization

Here we provide a mathematical model of the two-node temporal network construction, which is a stochastic process $\{S_m\}_{m\geq 0}$ coupling the activity of every unit. Complete mathematical details about the existence of $\{S_m\}_{m\geq 0}$ and the modeling of other systems are provided in the Supplementary Information.

We follow the notation in the Two-node systems section. According to the constraint $Z_n = X_n Y_n$, we construct a stochastic process $\{S_m\}_{m\geq 0}$, which for arbitrary sets of $t_1,...,t_k \in \mathbb{N}, k \in \mathbb{Z}^+$ satisfies

$$
\begin{aligned}
\mu_{S_{2t_1},...,S_{2t_k}} &= \mu_{X_{t_1},...,X_{t_k}}, \\
\mu_{S_{2t_1+1},...,S_{2t_k+1}} &= \mu_{Y_{t_1},...,Y_{t_k}}, \\
\mu_{(S_{2t_1}\cdot S_{2t_1+1}),...,(S_{2t_k}\cdot S_{2t_k+1})} &= \mu_{Z_{t_1},...,Z_{t_k}},
\end{aligned}
\tag{9}
$$

and $S_0 = S_1 = 1$. Here $\mu_{X_{t_1},...,X_{t_k}}$ represents the finite dimensional distribution of $\{S_m\}_{m\geq 0}$ at the time slice $(t_1,...,t_k)$. $\{S_m\}_{m\geq 0}$ can be viewed as composing of the following sequence

$$
(S_0, S_1,...,S_{2n}, S_{2n+1},...) = (X_0, Y_0,...,X_n, Y_n,...),
\tag{10}
$$

and $Z_n = S_{2n} S_{2n+1}$. $\{S_m\}_{m\geq 0}$ follows the activation order in the main text (i.e., the state of $x$ is determined first). If we exchange the order, the corresponding indexes in Eq. (9) and the sequence of $X_n$ and $Y_n$ in Eq. (10) are also swapped.

### System consistency

The consistency of the two-node system is equivalent to the existence of $\{S_m\}_{m\geq 0}$. If $\{S_m\}_{m\geq 0}$ is well-defined, the algorithmically produced IET distributions of $x, y, z$ will satisfy the target distributions $F, G, H$.

We derive the equivalence of the consistency condition for bursty and Poisson-like activity patterns in a two-node system. Let $p_x^{(n)} = p_x(\mathbf{0}^{(n-1)}, 1)$ denote the conditional probability that $x$ is active for the first time at $n$, where

$$
\mathbf{0}^{(m-1)} = \left(1, \underbrace{0,...,0}_{m-1}\right)^\mathsf{T}
\tag{11}
$$

is a trajectory with length $m$ and only one active state occurring at the initial time. For a power-law distribution with exponent $\alpha$,

$$
p_x^{(n)} \approx 1 - \left(\frac{n+\frac{1}{2}}{n-\frac{1}{2}}\right)^{-\alpha+1},
\tag{12}
$$

and for a discrete exponential distribution with exponent $\alpha$,

$$
p_x^{(n)} = 1 - e^{-\alpha}.
\tag{13}
$$

From the definition of the distribution consistency, the equivalence is given as

$$
\alpha_{node} \geq \alpha_{link}, p_x^{(1)} + p_y^{(2)} < 1, p_x^{(1)} + p_y^{(1)} < 1 + p_z^{(1)}
\tag{14}
$$

for bursty activity patterns and

$$
0 \leq \alpha_{node} - \alpha_{link} \leq \ln 2
\tag{15}
$$

for Poisson-like activity patterns. Note that these consistency conditions ensure that the algorithm works for any length $t_{\text{tol}}$.

### Construction of empirical temporal networks

We generate an unweighted underlying topology $\mathcal{S}$ and a temporal network $\mathcal{T}$ for each empirical dataset. We first determine the length of $\mathcal{T}$ by counting the number of timestamps in the dataset. Then, the snapshot at time $t$ is formed by all contact events with the corresponding timestamps. Finally, we obtain $\mathcal{S}$ by aggregating all snapshots, that is, link $(i, j)$ exists on $\mathcal{S}$ if individuals $i$ and $j$ interact at least once.

## Data availability

All the empirical datasets used in this paper are freely and publicly available at http://www.sociopatterns.org.

## Code availability

Code has been deposited into the publicly available GitHub repository at https://github.com/anzhisheng/Temporal-networks-by-spanning-trees[68].

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

## Acknowledgements

A.S. acknowledges support from the China Scholarship Council (no. 202206010147). A.L. and L.W. acknowledge support from the National Key Research and Development Program of China (no. 2022YFA1008400), National Natural Science Foundation of China (no. 62036002 and 62173004), and the Beijing Nova Program (Z211100002121105). Q.S. acknowledges support from Shanghai Pujiang Program (no. 23PJ1405500). J.B.P. acknowledges support from the John Templeton Foundation (grant number 62281).

## Author contributions

Conceptualization: A.S., A.L., L.W., and J.B.P.; Methodology: A.S.; Analysis and interpretation of data: A.S., Q.S., A.L., L.W., and J.B.P.; Writing the manuscript: A.S., Q.S., A.L., L.W., and J.B.P.

## Competing interests

The authors declare no competing interests.
