## [Peer Review File · Nature Communications]

REVIEWER COMMENTS

Reviewer #1 (Remarks to the Author):

The authors present a model to generate networks time-varying dynamics that follow desired inter-event time (IET) distributions. They consider scenarios where the system is constrained by an underlying fixed topology or when it evolves in time starting from an initial core of nodes (as in the classic BA model). I found the article well written (though as I detailed below some key parts would benefit from some clarification), the problem is interesting as the proposed solution. I have however some major concerns about the work presented and the details analysed.

The main point is the following. The authors limit the characterisation of the output of their model to the study of IET distributions, which anyway are put by hand and are not “emergent” from some simple mechanism. Burstiness is just one of the properties of real time-varying networks. There are complex temporal dynamics, which now have been shown to go beyond pairwise interactions. To be able to judge the model proposed we would need to investigate which types of structures/topologies the model generates at each time step, and what happens by integrating them in time. Clearly, with a static underlying structure the model will eventually return that structure, but what about at each time step? How do we arrive there? What about the weights (repetition of contacts)? The authors focus only on the burstiness, which is interesting, but at the end of day the model proposed is shown “just” to reproduce the IET distribution that are imposed by hand. For other people to use this model in applications across domains such as for example study diffusion/contagion processes, it would be important to get a feeling about which type of temporal topologies are generated by the model in terms of temporal topology. Are these realistic? This point is also very relevant for the analysis of real time-varying networks that authors conducted. Those real temporal networks also provide the sequence of links and nodes besides the IET distributions, does the model reproduce the structure of the networks (hence other key properties) besides the burstiness (which again is added by hand)?

While it is true that models that focus on real IET distributions are rare, there are some. For example, example the work by Burioni and colleagues Scientific Reports volume 7, Article number: 46225 (2017) is an example. That model extends the activity-driven framework (which is cited by the authors).

One of the assumptions in the model is that x , y , z need to be all active to result in a link. This assumption holds only on undirected network where acts are bidirectional. In directed communication networks for example (sending an email, or a message) is directional hence this does not hold necessarily. I suggest the authors to clarify this point.

Authors use sub and up scripts for the time step n (like in equation $n-1$). The difference between the two is rather unclear to me.

Eq 3 is rather important in the construction. Its description is not clear to me. I would suggest spending a bit more time going through the details.

In section 2.2 authors speak about “layers”. However, in the context considered here the word layer is misleading. I would suggest speaking about nodes at two steps, or shells?

What are the implications of the assumption described in line 143-145 on correlated dynamics which have more and more identified to be very important on time-varying networks?

In section 2.3 some of the details of the final models are not clear and left on other parts of the submitted materials. For example, more details about how the root and the elements in the spanning tree are selected should be given in the main.

The fact that the model returns the imposed IET looks to me more a sanity check for consistency rather than a result. The authors should articulate a bit more about this point.

The authors consider an evolving network in the final part. However, some systems do not increase in size, just the link evolve as function of time. Their construction following the BA dynamics seems incompatible with these cases. On a similar point, the arrival of a node per time is rather unrealistic, the BA graph is popular because what happens when these dynamics are integrated in time, in the case of time-varying networks this is not the case. The temporal dynamics taking place at a particular time-scale are the key. Here, we do not have a real discussion about such dynamics.

The authors speak about link removal, ageing etc.. These details are left completely to the SM. I would suggest the authors to discuss them in the main as well.

Reviewer #2 (Remarks to the Author):

In this manuscript, the authors present a method for producing a temporal network (in discrete time) in which the inter-event times for links and nodes follow desired distributions. The main idea is that one can impose the IET distribution on a first node and its links and neighbours by treating them as independent 2-nodes systems, and then to propagate this along the underlying static network, assumed to be known (it would be e.g. the aggregated network structure), using a spanning tree for the propagation. The activity on links not

belonging to the spanning tree is then dictated by the simultaneous activity of its extremity nodes. This is possible under some condition of consistency between the IET of nodes and links, and of conditional independence between the activity of nodes at distance 2.

The authors present the method, and then show it at work in several examples. This manuscript and method can be of interest to scientists working on temporal networks, however the potential audience is rather limited, and this is a technical article that would be much more suited to a more specialized journal after some revision. In particular:

-the title is very misleading: it includes the word "emergence"; however, what is described is an ad-hoc method to create a network with desired properties by putting these properties by hand: this is exactly the opposite of the concept of an emerging phenomenon

-the authors mention several temporal network models, then write that "they usually fail to reproduce the level of burstiness as empirical datasets". This is incorrect, as several of the models of these references do present bursty behaviour. Note that there are some other models that do the same such as Ubaldi et al, Sci Rep 2017, Karsai et al. Sci Rep 2012, Lebail et al. Phys Rep 2023.

-the fundamental procedure is to put by hand desired IET. In this respect the procedure is not fundamentally different from other proposals to create temporal networks with ad hoc properties by

imposing distributions and creating renewal processes. The novelty here is thus interesting from a methodological point of view but rather limited in scope.

-the procedure is rather clear for the spanning tree. However, whether/how the IET on branches (links not part of the spanning tree) are satisfied is quite unclear. The authors write " when a tree system is consistent, the algorithmic IET distribution of every single node/link fulfills its corresponding target distribution (see Supplementary Material section 2).", however this is not explained in the SM, just claimed: "We claim that the system is consistent if and only if its spanning tree system is consistent. In particular, when all nodes and trunks have the same targeted IET distribution, the whole system is synchronous, and therefore all nodes and links (including branches) always become active or inactive at the same time". In particular, it is quite puzzling to obtain a totally synchronous system as this is not realistic at all.

Reviewer #3 (Remarks to the Author):

The authors present a framework for generating temporal networks with activity patterns fit to a target inter-event time distribution. The method operates by fitting activation probabilities on a rooted tree. Graphs with cycles are treated by extracting a spanning tree and applying the method to that tree first, then extending activation probabilities to edges that do not belong to that tree in a straightforward manner. The method has several assumptions that are encoded in equation 3 and relate to consistency criteria for node-level and edge-level activation, as well as independence criteria. The method is demonstrated using synthetic and empirical temporal networks and good fits for activity distributions are obtained for various topologies.

Overall, the paper is well-written and the methods are carefully justified. I have several comments related to clarity of the presentation and interpretation of the results. I have roughly ordered my comments from most major to most minor.

The assumption that an edge's activity is completely defined by the activity of its endpoints seems quite restrictive to me. For example, in a temporal network, four individuals may be part of a clique, and two pairs may interact without all four simultaneously interacting. Is it correct to say that such a thing cannot happen in the model presented here? If so, how severely does this impact our ability to interpret the output of the algorithm as a temporal network? In other words, do we obtain a good fit for

activation frequencies that is a poor representation of other features of the underlying dynamical processes?

In figure 3, the "algorithm" data points are consistently above the "empirical" data points. Perhaps the authors can briefly comment on why this is the case? Is this an expected outcome when using the fitting approach employed here, or is this just a coincidence?

In the main text, it is mentioned that figure S5 demonstrates robustness to spanning tree selection. From the caption of that figure, however, it is not clear how much the spanning tree is varied, as only the root node selection is discussed. In principle, it is possible that the same spanning tree is selected for many (or in the worst case, all) roots; this will depend on how a spanning tree is generated from a specified root.

Related to the previous point, figure S5 demonstrates a kind of macro-scale robustness, but I wonder if there is less robustness on the micro-scale. For example, is there any observable difference between the activity patterns of trunk edges and branch edges?

In figures 2, 3, and 4B, the horizontal axes for the insets do not match the scales for the main panels. Furthermore, from the insets, it appears that data points are cut off in the main panels (though it is hard to be sure because the scales do not match).

In the discussion, the authors write "Our analysis of activity patterns that are robust to underlying topology indicates that activity patterns of a network system is strongly determined by the dynamics of its spanning tree." I think it is a stretch to claim that the dynamics are determined by a spanning tree. In part, this is because spanning trees are not (in general) unique. Also, and more crucially, the authors only demonstrate that a spanning tree is sufficient to fit a distribution of activity patterns; this does not mean that they have captured the underlying causal mechanisms, or specific local features of activity, which may both depend strongly on the cycle structure, for example.

The authors use the phrase "robust to underlying topology" in several places to describe the activity patterns. I feel this phrasing is somewhat strange, as it seems to actually refer to the robustness of the algorithm's ability to fit a target activity pattern, not the robustness of the pattern itself. Naively, I would expect it to mean that the parameters obtained in the fitting procedure do not strongly depend on the underlying topology, but I don't think that is the intended meaning.

There is a small typo in the caption of figure 1: "constraint" should be "constrain".

Line 300 of the supporting material has a typo: "bounded" should be "bound".

The authors may wish to consider making the software used in the analysis available in some form. Algorithms 1 & 2 provide enough information for someone to write their own implementation anyway, so it seems like there is little to lose by doing so.

Response to Reviewer #1:

The authors present a model to generate networks time-varying dynamics that follow desired inter-event time (IET) distributions. They consider scenarios where the system is constrained by an underlying fixed topology or when it evolves in time starting from an initial core of nodes (as in the classic BA model). I found the article well written (though as I detailed below some key parts would benefit from some clarification), the problem is interesting as the proposed solution. I have however some major concerns about the work presented and the details analysed.

We are grateful to the reviewer for their excellent summary of the manuscript and constructive comments for its improvement. We have followed all of the reviewer's suggestions, undertaking new research and revising our manuscript.

The main point is the following. The authors limit the characterisation of the output of their model to the study of IET distributions, which anyway are put by hand and are not "emergent" from some simple mechanism.

We agree: the entire purpose of our study is to develop a method to construct temporal networks that are guaranteed to match, as much as possible, any desired IET distribution. The resulting IET that arises from our algorithm is not an "emergent" property, but rather one constructed by design.

We have removed the term "emergent" altogether, and clarified that our goal is to develop a method of constructing any desired IET distribution for nodes and edges – a problem that is non-trivial, and not previously solved.

Burstiness is just one of the properties of real time-varying networks. There are complex temporal dynamics, which now have been shown to go beyond pairwise interactions. To be able to judge the model proposed we would need to investigate which types of structures/topologies the model generates at each time step, and what happens by integrating them in time. Clearly, with a static underlying structure the model will eventually return that structure, but what about at each time step? How do we arrive there? What about the weights (repetition of contacts)? The authors focus only on the burstiness, which is interesting, but at the end of day the model proposed is shown "just" to reproduce the IET distribution that are imposed by hand.

We agree with the critique. We have focused on constructing a method to produce any given IET distribution. Although we have focused on the burstiness of the IET distribution (at least in our title), we emphasize that we can actually construct any desired IET distribution, recapitulating the entire probability distribution, not just the properties of its tails (that is, whether it is bursty or not). We emphasize that "just" doing this "by hand" is already a highly non-trivial problem that has resisted solution despite prior attempts.

Figure R1: Weighted node degree of aggregated networks. (A) We construct the aggregated network which collects all interactions from the first snapshot to the t_{agg} th snapshot. (B-C) The weighted node degree distribution exhibits a stationary distribution for different aggregation times t_{agg} in scale-free (B) and small world underlying topologies (C), which is consistent with our theoretical prediction (see Equation 8 in the main text).

We agree that the question about weights (repetition of contact) is also very interesting, and it differs from the IET distribution. We address this question below.

For other people to use this model in applications across domains such as for example study diffusion/contagion processes, it would be important to get a feeling about which type of temporal topologies are generated by the model in terms of temporal topology. Are these realistic? This point is also very relevant for the analysis of real time-varying networks that authors conducted. Those real temporal networks also provide the sequence of links and nodes besides the IET distributions, does the model reproduce the structure of the networks (hence other key properties) besides the burstiness (which again is added by hand)?

Yes, there are many ways to summarize a sequence of active links and nodes in a temporal network. Inter-event time distribution (IET) is by far the most common way to quantify temporal dynamics, and this is why we have focused on the IET in our study. But indeed there are many other ways to summarize temporal dynamics, distinct from the IET.

In fact, we do also study other aspects of the temporal network, not just the IET. Namely, we also study the weighted node degree distribution obtained by aggregating the state of the network across many timepoints (that is, the repetition of contacts, which the referee

suggested, see Figure R1A). We provide a theoretical prediction for this distribution (see Equation 8 in the main text) and we verify it on synthetic temporal networks (see Figures R1B and R1C). Our algorithm also does a good job at reproducing the stationary property of the weighted node degree distribution observed in empirical networks (see Figure R2), as well as (simultaneously) matching the IET.

And so, in summary, our model reproduces both empirical inter-event times as well as another important summary of the dynamic networks – namely, the stationary weighted degree of each node.

Figure R2: Matching the weighted node degree distributions of empirical temporal networks. Our construction also reproduces the stationary weighted node degree distributions in empirical networks. (A) empirical results; (B) algorithmic results.

While it is true that models that focus on real IET distributions are rare, there are some. For example, example the work by Burioni and colleagues Scientific Reports volume 7, Article number: 46225 (2017) is an example. That model extends the activity-driven framework (which is cited by the authors).

Thank you for pointing us to this important paper, which does indeed discuss the long-tails (burstiness) of empirical IET distributions. Nonetheless, the model and construction developed by Burioni et al consider only the burstiness of single individuals, assuming the activities of individuals are independent of each other. But in empirical datasets, not only do individual nodes show bursty activity patterns, but so do links between individuals show

burstiness (see Figure 3 in the main text). In our study, by contrast to Burioni et al, we manage to simultaneously reproduce the burstiness of nodes and edges observed in empirical datasets. This is a substantially more difficult problem because it requires that the activity of individuals is governed by the activity of mutual links, and therefore not independent.

Thank you again for pointing us to the work of Burioni and colleagues, which we now cite and explain how it differs from our own study.

One of the assumptions in the model is that x , y , z need to be all active to result in a link. This assumption holds only on undirected network where acts are bidirectional. In directed communication networks for example (sending an email, or a message) is directional hence this does not hold necessarily. I suggest the authors to clarify this point.

We thank the reviewer for this great suggestion. The reviewer is correct: one of the assumptions of our model is that the underlying topology is undirected. In this case, the activity of every single node and edge is highly related, because a link is active only when the two nodes on both sides of the link are active. The main advance of our study is to find a model that coordinates the activity of single nodes and edges in the underlying topology, such that the output IET distributions of single nodes and edges match the targeted distributions.

However, as the referee says, for directed links the occurrence of an interaction (such as sending an email) totally depends on the sender. In this case, the activity of nodes is independent of each other. A model for constructing such dynamic patterns to match empirical data is far more simple, and already exists in the literature, eg Barrat A, Fernandez B, Lin K K, et al. Physical Review Letters 110 (2013), which we have cited in the manuscript.

Authors use sub and up scripts for the time step n (like in equation n 1). The difference between the two is rather unclear to me.

Thanks for catching this, we have revised the text to clarify the notation. For a node x , we use X_n to denote a random variable that records the state of x at time n . By contrast, $\mathbf{X}^{(n)}$ is a random vector of length $n + 1$, given by $(X_0, X_1, \dots, X_n)^T$, which records the entire history of states of X from time 0 to time n .

Eq 3 is rather important in the construction. Its description is not clear to me. I would suggest spending a bit more time going through the details.

Thanks for the suggestion, because this is a key equation. In the revised manuscript, we have provided a more detailed form of Equation 3 and explained the meaning of the various terms. In brief: the first probability p_1 denotes the probability of link z being active given that nodes x and y are both active. The last three probabilities all denote the probability of z being inactive. The difference between these terms gives p_2 (respectively p_3 or p_4) – the chance that x is active but y is inactive (or, receptively, y is active but x is inactive, or both x and y are inactive).

In section 2.2 authors speak about “layers”. However, in the context considered here the word layer is misleading. I would suggest speaking about nodes at two steps, or shells?

Agreed, We have followed the reviewer’s suggestion to change the word “layer” terminology explaining more clearly, such as “two steps away from the root”.

What are the implications of the assumption described in line 143-145 on correlated dynamics which have more and more identified to be very important on time-varying networks?

This is an excellent question. As mentioned above, the activity of nodes and links are coupled with each other because of the existence of underlying topologies, so it is a challenge to coordinate the activity of single nodes and links. In our study, we provide a minimal model that the activity of a node x which is l step away from the root of the spanning tree only depends on the state of the node y , which is connected to x and $l - 1$ step away from the root, and we determine the state of x via the two-node system formed by (x, y) . We call this the one-step effect. In the real world, this effect implies that the activity of individuals is only related to their first-order neighbors.

In section 2.3 some of the details of the final models are not clear and left on other parts of the submitted materials. For example, more details about how the root and the elements in the spanning tree are selected should be given in the main.

Thank you for this suggestion. In our model, the selection of the spanning tree and the root of the tree is random. The choice of root does not affect the output IET distributions of nodes and edges, and we also show that the output is stable for the selection of spanning tree in the SM (see Figure S5). We have modified the text to clarify this in the revision.

The fact that the model returns the imposed IET looks to me more a sanity check for consistency rather than a result. The authors should articulate a bit more about this point.

We disagree. The fact that the model returns the target IET is in fact the primary result: we have provided a method to construct a temporal network that matches any (self-consistent) inter-event time distribution.

The purpose of these figures is to show, graphically, that our construction actually works – and so, yes, it is a sanity check in that sense. But we emphasize that no other constructive method exists in the literature that can produce dynamic networks that match empirical (or other target) IETs. And so showing that our construction actually works is a key result.

The authors consider an evolving network in the final part. However, some systems do not increase in size, just the link evolve as function of time. Their construction following the BA dynamics seems incompatible with these cases. On a similar point, the arrival of a node per time is rather unrealistic, the BA graph is popular because what happens when these dynamics are integrated in time, in the case of time-varying networks this is not the case. The

temporal dynamics taking place at a particular time-scale are the key. Here, we do not have a real discussion about such dynamics.

We agree with the reviewer and we have undertaken new research to address this important point.

Aside from overall system growth, there can be time-varying underlying topologies caused by the evolution of the links alone. And so in the revision, we consider a time-varying underlying topology with a periodic transition between three small-world networks of the same size $N = 20$ and average degree $k = 6$ (see Figure R3A). In this case, these three networks have the same number of links but different topologies. We show that our model can also successfully generate bursty activity patterns on this time-varying underlying topology (see Figure R3B). We added text describing this new result, including the additional Figure R3, which is now Figure S10 in the revised manuscript.

Figure R3: Burstiness on a time-varying underlying topology with periodic transition. (A) The three networks are the underlying topologies at the corresponding time steps with the same number of edges. (B) Our construction successfully generates the targeted burstiness in this time-varying underlying topology.

We also agree with the reviewer that, in cases when the number of nodes grows over time, the number of nodes introduced need not be constant at each time step and it is usually larger than 1 on average. In our study, we use the BA dynamics as an example to test the performance of our model when the size of underlying topologies is growing. The result shows that our model can deal with the case of non-constant rates of growth. Besides, the BA dynamics can be seen as a high-resolution detection, where any growth of system size will be recorded. Based on the BA dynamics, an approach to model multiple added nodes at a single time step is to aggregate multiple snapshots together to form a new single snapshot. We undertook similar simulations based on the above case, where our algorithm is also applicable.

The authors speak about link removal, ageing etc.. These details are left completely to the SM. I would suggest the authors to discuss them in the main as well.

In the revised manuscript, we discuss a broader range of time-varying underlying topologies induced by network transitions and link removal, in the main text.

Response to Reviewer #2:

In this manuscript, the authors present a method for producing a temporal network (in discrete time) in which the inter-event times for links and nodes follow desired distributions. The main idea is that one can impose the IET distribution on a first node and its links and neighbours by treating them as independent 2-nodes systems, and then to propagate this along the underlying static network, assumed to be known (it would be e.g. the aggregated network structure), using a spanning tree for the propagation. The activity on links not belonging to the spanning tree is then dictated by the simultaneous activity of its extremity nodes. This is possible under some condition of consistency between the IET of nodes and links, and of conditional independence between the activity of nodes at distance 2.

We thank the reviewer for reviewing our manuscript carefully and providing constructive comments. Please find below our detailed responses to each comment.

The authors present the method, and then show it at work in several examples. This manuscript and method can be of interest to scientists working on temporal networks, however the potential audience is rather limited, and this is a technical article that would be much more suited to a more specialized journal after some revision. In particular:

-the title is very misleading: it includes the word "emergence"; however, what is described is an ad-hoc method to create a network with desired properties by putting these properties by hand: this is exactly the opposite of the concept of an emerging phenomenon

We entirely agree about the title: the word emergent is completely wrong. Indeed, the fact that we are able to match a target IET distribution (including the bursty IETs seen in empirical data), is due directly to the constructive method we develop, as opposed to some mysterious emergent property. We have changed our title to "Constructing temporal networks with bursty activity patterns" in the revision.

-the authors mention several temporal network models, then write that "they usually fail to reproduce the level of burstiness as empirical datasets". This is incorrect, as several of the models of these references do present bursty behaviour. Note that there are some other models that do the same such as Ubaldi et al, Sci Rep 2017, Karsai et al. Sci Rep 2012, Lebail et al. Phys Rep 2023.

Thank you for pointing us to these papers, which we now cite, discuss, and compare to our results in the revised manuscript.

There is a key difference between our results and those produced by these three prior studies. Although all the studies are concerned with bursty activity patterns seen in empirical data, only our model is capable of reproducing the full extent of empirical patterns. In particular, although some of the cited papers can produce burstiness in the node activities, they do not simultaneously reproduce the burstiness of edge activities observed in empirical

data. And so our results provide a substantial advance over these prior studies.

Moreover, as shown in Figure 3 in the main text, the exponent of the IET distribution for nodes is often different from that for edges, and the exponents for nodes and edges also differ widely across different empirical datasets. Our construction is able to produce dynamic networks that match the exponents of edges and nodes simultaneously, and do so across a wide variety of exponents observed in empirical data. By contrast, the methods proposed by Lebail et al and Karsai et al fail to achieve these results: a direct comparison is shown in Figure R4 below.

Figure R4: The performance of reproducing the simultaneous burstiness of inter-event times for nodes and edges in empirical datasets, by different models. We use the second empirical dataset (Office 13) to test the performance of the models. Ubaldi’s model (the first row) successfully reproduces the burstiness of nodes, but it fails to reproduce the burstiness of edges. Lebail’s model fails to reproduce the burstiness of both nodes and edges. Karsai’s model is applied to single-agent cases, so there is no activity of links. Our model can reproduce the same level of burstiness as the empirical dataset, in both nodes and edges.

As this figure shows, the model proposed by Ubaldi et al provides an extended version of the activity-driven model, but it focuses on node degree distribution for the aggregated

structure of a temporal network. The authors mention “burstiness”, but only for the activity of nodes. We implemented the Ubaldi model, shown in the first row of Figure R4. As advertised, the IET distribution of nodes produced by their model does satisfy a power-law distribution, and it can fit the empirical distribution of nodes; but the IET distribution of edges they produce is not a power-law distribution, and it deviates from the empirical distribution of edges. In fact, since the activity of nodes in their model is independent, the IET distribution of links is not mathematically guaranteed to be a power-law distribution.

For the model proposed by Lebail, the activity of nodes is the same as the original activity-driven model, which means the IET distribution of single nodes is an exponential distribution, rather than a heavy-tailed distribution. This means that the model of Lebail cannot produce the empirical burstiness of node activities, as seen in the second row of Figure R4.

For Karsai’s paper, the main contribution is to propose a new statistical measurement, the distribution of the number of events in a given time interval, to characterize temporal networks, rather than to discuss the behaviour of IET distributions. The generative model mentioned in the paper is applied to the case of single agents. It is not applied to multiple individuals, and therefore produces no pattern of activities of links whatsoever.

We thank the referee again for pointing out these prior papers. We have discussed them in the revised manuscript and clarified how our work significantly extends these prior studies.

-the fundamental procedure is to put by hand desired IET. In this respect the procedure is not fundamentally different from other proposals to create temporal networks with ad hoc properties by imposing distributions and creating renewal processes. The novelty here is thus interesting from a methodological point of view but rather limited in scope.

We basically agree: our main result is a method to produce any desired IET (of both nodes and edges both) “by hand”, using renewal theory. We believe this is an important and novel contribution. Even though inter-event times in temporal networks are widely studied across many fields, there is no prior solution to the most basic problem of how to produce temporal networks with the dynamic properties observed in empirical settings (i.e. the simultaneous burstiness of nodes and edges). And so our contribution is to provide a solution to this basic problem.

-the procedure is rather clear for the spanning tree. However, whether/how the IET on branches (links not part of the spanning tree) are satisfied is quite unclear. The authors write “ when a tree system is consistent, the algorithmic IET distribution of every single node/link fulfills its corresponding target distribution (see Supplementary Material section 2).”, however this is not explained in the SM, just claimed: “We claim that the system is consistent if and only if its spanning tree system is consistent.

This is a great question, thank you for asking us to clarify. In our model, we can only

assign IET distributions to the elements in the spanning tree (i.e. all nodes and trunks). Therefore, for a more general static underlying topology, we select a spanning tree and first determine the state of all nodes and trunks at each step time. Then the state of a branch is determined by the state of its two endpoints. Specifically, a branch is active if and only if its two endpoints are all active, which is the same as the setup for trunks. Although we cannot guarantee that the branches will match the target IET distribution, we have proved that the IET distribution of every branch is uniformly upper- and lower-bounded by heavy-tailed distributions or exponential distributions under bursty activity pattern or Poisson-like activity pattern (section 3 in the SM). This proof provides a theoretical guarantee that the IET distribution of nodes and edges is robust to underlying topologies and spanning tree selection. The corresponding numerical simulations are shown in Figures S3 and S5. We have clarified these points in the main text of the revision.

Some of the confusion here is because we mis-used the word “claim,” when we actually meant to write “define”. In fact, we define a general network as consistent if and only if its spanning tree system is consistent. For the two-node and tree underlying topologies, we derive a corresponding stochastic process for construction of temporal networks over these tree topologies, and we define that the two-node (or tree) system is consistent if and only if the corresponding stochastic process is well-defined. We use the Kolmogorov extension theorem to determine whether the stochastic process is well-defined, as described in sections 2.1.1 and 2.2.1 in the SM. As a result, if the system is consistent, our model mathematically guarantees that the algorithmic IET distribution of every node and link in the spanning tree will match the corresponding target IET distribution. When we consider a more general underlying topology (beyond just spanning trees), we first determine the activity of its spanning tree by Algorithm 2 in the SM, and then we decide the state of the elements outside of the tree (i.e. branches) by the rule discussed in the previous paragraph. This construction is summarized in Algorithm 3 in the SM.

In particular, when all nodes and trunks have the same targeted IET distribution, the whole system is synchronous, and therefore all nodes and links (including branches) always become active or inactive at the same time”. In particular, it is quite puzzling to obtain a totally synchronous system as this is not realistic at all.

This is correct: if we demand that all nodes and trunks have the same target IET distribution, then the system will be synchronous. And, moreover, we agree that this is an unrealistic case.

But importantly our construction allows us to have a different target IET for each node and each trunk. And so our construction works to produce realistic activity patterns that are not synchronous. In fact, almost all the figures shown in the paper exhibit cases without synchrony.

Response to Reviewer #3:

The authors present a framework for generating temporal networks with activity patterns fit to a target inter-event time distribution. The method operates by fitting activation probabilities on a rooted tree. Graphs with cycles are treated by extracting a spanning tree and applying the method to that tree first, then extending activation probabilities to edges that do not belong to that tree in a straightforward manner. The method has several assumptions that are encoded in equation 3 and relate to consistency criteria for node-level and edge-level activation, as well as independence criteria. The method is demonstrated using synthetic and empirical temporal networks and good fits for activity distributions are obtained for various topologies.

Overall, the paper is well-written and the methods are carefully justified. I have several comments related to clarity of the presentation and interpretation of the results. I have roughly ordered my comments from most major to most minor.

We are grateful to the reviewer for their careful reading of the manuscript, precise summary, their overall assessment, and their helpful suggestions for revision and research.

The assumption that an edge's activity is completely defined by the activity of its endpoints seems quite restrictive to me. For example, in a temporal network, four individuals may be part of a clique, and two pairs may interact without all four simultaneously interacting. Is it correct to say that such a thing cannot happen in the model presented here?

Agreed – that would be extremely restrictive. But our model is not restrictive in this way, and indeed a pair of individuals may interact without all four nodes interacting.

This can occur in our model because the underlying topology (that is, which edges are even possible for interaction) can change dynamically. In other words, not only is the pattern of active nodes/edges changing over time, but also the underlying topology of which edges are even allowed to become active can also change over time. As a result, there may be periods of time where all four individuals are in a clique – and all four must be interacting when active, in such states – as well as periods of time where only two of the four nodes are connected in the underlying topology, and those two alone can be interacting without the other two.

Motivated by the reviewer's comment, we have revised the terminology in the manuscript to explain more clearly what it means for the "underlying topology" (that is, which edges are even possible for interactions) to change over time. In particular, we clarify two easily confused concepts: "time-varying underlying topologies" and "temporal networks". The underlying topology represents the physical limitation on pairwise interactions, where an interaction between individuals i and j has a chance to occur only when there is a link between i and j on the underlying topology at that time. Whereas the temporal network records the

actual activity of nodes x , y , and the link between them. When the underlying topology is time-varying, we first specify the underlying topology in each moment, then we determine the activity of the nodes and links on that topology according to our algorithmic construction from renewal theory.

In our revised manuscript, we show that our method can deal with time-varying underlying topologies, which accommodates the situation the referee describes. To illustrate this, we can consider a time-varying underlying topology with periodic transitions between the five networks shown in Figure R5, which models the scenario the referee describes above. (It is also possible to extend our model to random transitions between underlying topologies). This is a case where a common clique of four individuals can pass periods when only two (or three) members need to be active simultaneously.

Figure R5: Transitions in the underlying topology for four nodes. A completely connected clique of size 4 can periodically transition into distinct cliques of size 2. Depending upon the underlying topology at a given time point, all five edges must be active when all nodes are active (network 1), or two or four edges can be active when all nodes are active (networks 2-5). Our construction successfully generates the targeted IET distribution, even when underlying topology is time-varying in this way.

In figure 3, the “algorithm” data points are consistently above the “empirical” data points. Perhaps the authors can briefly comment on why this is the case? Is this an expected outcome when using the fitting approach employed here, or is this just a coincidence?

Thank you for pointing this out. The reason for this phenomenon is that the empirical IET distributions for nodes and edges all follow heavy-tailed distribution, but they are not strictly power-law distributions. In our model, we use a true power law to fit these IET distributions, and so there are some slight differences between the empirical and algorithmic results. Specifically for Figure 3, a standard power-law distribution (i.e. the algorithmic IET distributions) is a straight line in log-log coordinates, while the empirical data are not. This result in a slight verticle offset between the data and the (true power-law) model – but the key point is that the best-fit slope matches the data very well, meaning that the degree of burstiness is matched.

In the main text, it is mentioned that figure S5 demonstrates robustness to spanning tree selection. From the caption of that figure, however, it is not clear how much the spanning tree is varied, as only the root node selection is discussed. In principle, it is possible that the same spanning tree is selected for many (or in the worst case, all) roots; this will depend on how a spanning tree is generated from a specified root.

This is a reasonable concern, thank you for asking. In fact, the spanning trees vary considerably as we choose different root nodes. In particular, there is no isomorphism between any pair of spanning trees that we have selected – they are all distinct. We have clarified this important point in the revision, and added a new figure to show that all the spanning trees are distinct (Figure S5A).

Related to the previous point, figure S5 demonstrates a kind of macro-scale robustness, but I wonder if there is less robustness on the micro-scale. For example, is there any observable difference between the activity patterns of trunk edges and branch edges?

This is a good point – indeed, there is a difference between the activity of trunks and branches. In Figure S1 in the SM, we plot the IET distribution of every single node, trunk, and branch. Fluctuations for links are more substantial than that for nodes, because the activity of trunks and branches are different. We can classify all branches according to the size of the ring that they belong to. We then plot the IET distribution of the branches located in the largest and smallest size ring. The result shows that the exponent of the IET distribution is lower when a branch in a larger size ring. And so we can partly understand the source of this variation between trunk and branch.

Despite for this difference, we have proven that the IET distribution of any branch still follows a heavy-tailed distribution and it is uniformly upper- and lower-bounded by two power-law distributions (section 3.2 in the SM). This result guarantees the macro-scale stability associated with spanning tree selection.

In figures 2, 3, and 4B, the horizontal axes for the insets do not match the scales for the main panels. Furthermore, from the insets, it appears that data points are cut off in the main panels (though it is hard to be sure because the scales do not match).

Thank you for pointing this out. We have redrawn these figures to ensure that the horizontal axes for the inset and main panel are matched, and have carefully checked that there is no cutoff data point in these figures.

In the discussion, the authors write “Our analysis of activity patterns that are robust to underlying topology indicates that activity patterns of a network system is strongly determined by the dynamics of its spanning tree.” I think it is a stretch to claim that the dynamics are determined by a spanning tree. In part, this is because spanning trees are not (in general) unique. Also, and more crucially, the authors only demonstrate that a spanning tree is suffi-

cient to fit a distribution of activity patterns; this does not mean that they have captured the underlying causal mechanisms, or specific local features of activity, which may both depend strongly on the cycle structure, for example.

We mostly agree with these points, and we have revised them accordingly. We agree that some local features cannot be captured by the activity of a spanning tree, as we now explain in the revised discussion. At the same time, however, we have truly shown that macroscopic features are not dependent on the choice of spanning tree and are largely determined by (any) spanning tree.

The authors use the phrase "robust to underlying topology" in several places to describe the activity patterns. I feel this phrasing is somewhat strange, as it seems to actually refer to the robustness of the algorithm's ability to fit a target activity pattern, not the robustness of the pattern itself. Naively, I would expect it to mean that the parameters obtained in the fitting procedure do not strongly depend on the underlying topology, but I don't think that is the intended meaning.

Thanks for pointing out the possible ambiguity. We have followed the reviewer's suggestion and have changed that to say "The algorithmic IET distribution of nodes or edges does not strongly depend on the underlying topology".

There is a small typo in the caption of figure 1: "constraint" should be "constrain".

Corrected.

Line 300 of the supporting material has a typo: "bounded" should be "bound".

Corrected.

The authors may wish to consider making the software used in the analysis available in some form. Algorithms 1 & 2 provide enough information for someone to write their own implementation anyway, so it seems like there is little to lose by doing so.

Great idea: we have produced a GitHub repository at <https://github.com/anzhisheng/Temporal-networks-by-spanning-trees>, where we have uploaded the source code for producing both bursty activity patterns and Poisson-like activity patterns.

REVIEWERS' COMMENTS

Reviewer #1 (Remarks to the Author):

I would like to thank the authors for the great work done in revising the article. All the unclear points have been nicely addressed. I commend their responses to all the comments and criticisms raised by me and the other two referees. Well done.

I am happy to see that some of the suggestions I made helped expanding the article. I am referring in particular to the study of the weight distributions. I am however a bit surprised that the authors did not investigate the weight distributions of the real networks and compared them with those resulting from their model. Showing an agreement in this direction would have been a particularly strong result.

Overall, I think the article is now in a much better shape. However, I share the sentiment of at least one of the other referees and feel that the work presented here should be probably published in a more technical journal.

I believe the authors did not present a convincing argument about the relevance of their effort across domains. For a publication in a journal like this one the author should not only present the model but also show its importance/relevance in practical problems. I would suggest the author to consider investigating the impact of these dynamics on different types of contagion processes. We know burstiness is particularly important and that neglecting it leads to misrepresentations of processes. Furthermore, the study of the weights is very interesting but I think more effort should go in that direction, for example, considering real networks.

Reviewer #2 (Remarks to the Author):

The authors have convincingly answered my comments and revised their manuscript. In addition they now make their code available. This article and method could thus interest many researchers working on temporal networks and interested in using synthetic networks with prescribed distributions of inter-event times. In addition it could stimulate further modeling work in this direction.

For these reasons, I think this article can now be published in Nature Communications.

Reviewer #3 (Remarks to the Author):

The authors have addressed my prior comments to my satisfaction.

I only add a few extremely small points that the authors may address at their discretion:

- Lines 57, 235: it is a bit odd to start a sentence with "And".
- Line 94: the addition could be read as suggesting future work, rather than foreshadowing results that are presented later in the manuscript.
- Line 153: "two-nodes" should be "two-node"

Response to Reviewer #1:

I would like to thank the authors for the great work done in revising the article. All the unclear points have been nicely addressed. I commend their responses to all the comments and criticisms raised by me and the other two referees. Well done.

We thank the referee for all their constructive input, and we are happy that the unclear points are now nicely addressed.

I am happy to see that some of the suggestions I made helped expanding the article. I am referring in particular to the study of the weight distributions. I am however a bit surprised that the authors did not investigate the weight distributions of the real networks and compared them with those resulting from their model. Showing an agreement in this direction would have been a particularly strong result.

In fact we do show a comparison between the weight distribution of the four empirical networks and the corresponding distribution for networks generated by our spanning tree algorithm (Supplementary Fig. S12).

Overall, I think the article is now in a much better shape. However, I share the sentiment of at least one of the other referees and feel that the work presented here should be probably published in a more technical journal.

I believe the authors did not present a convincing argument about the relevance of their effort across domains. For a publication in a journal like this one the author should not only present the model but also show its importance/relevance in practical problems. I would suggest the author to consider investigating the impact of these dynamics on different types of contagion processes. We know burstiness is particularly important and that neglecting it leads to misrepresentations of processes. Furthermore, the study of the weights is very interesting but I think more effort should go in that direction, for example, considering real networks.

As mentioned above, the manuscript does indeed include the weight distributions on real temporal networks, compared to those distributions produced by our algorithm; they show strong agreement (Supplementary Fig. S12).

Response to Reviewer #2:

The authors have convincingly answered my comments and revised their manuscript. In addition they now make their code available. This article and method could thus interest many researchers working on temporal networks and interested in using synthetic networks with prescribed distributions of inter-event times. In addition it could stimulate further modeling work in this direction. For these reasons, I think this article can now be published in Nature Communications.

We thank the referee for all their constructive input, and for assessing the value of the study more broadly.

Response to Reviewer #3:

The authors have addressed my prior comments to my satisfaction.

We thank the referee for all their constructive input and feedback.

I only add a few extremely small points that the authors may address at their discretion:

- Lines 57, 235: it is a bit odd to start a sentence with "And".

- Line 94: the addition could be read as suggesting future work, rather than foreshadowing results that are presented later in the manuscript.

- Line 153: "two-nodes" should be "two-node"

We have fixed these errors in the final version of the manuscript file.